# Betraying Blockchain: Accountability, Transparency and Document Standards for Non-Fungible Tokens (NFTs)

**Kristin Cornelius**

Center for Information as Evidence, University of Los Angeles, Los Angeles, CA 90095, USA; krisbcorn@ucla.edu

**Abstract:** Transparency and accountability are important aspects to any technological endeavor and are popular topics of research as many everyday items have become 'smart' and interact with user data on a regular basis. Recent technologies such as blockchain tout these traits through the design of their infrastructure and their ability as recordkeeping mechanisms. This project analyzes and compares records produced by non-fungible tokens (NFTs), an increasingly popular blockchain application for recording and trading digital assets, and compares them to 'document standards,' an interdisciplinary method of contract law, diplomatics, document/interface theory, and evidentiary proof, to see if they live up to the bar that has been set by a body of literature concerned with authentic documents. Through a close reading of the current policies on transparency (i.e., CCPA, GDPR), compliance and recordkeeping (i.e., FCPA, SOX, UETA), and the consideration of blockchain records as user-facing interfaces, this study draws the conclusion that without an effort to design these records with these various concerns in mind and from the perspectives of all three stakeholders (Users, Firms, and Regulators), any transparency will only be illusory and could serve the opposite purpose for bad actors if not resolved.

**Keywords:** blockchain; records; non-fungible tokens; transparency





## 1. Introduction

Broadly, transparency and accountability are current topics of interest. Researchers from several fields, regulators from various sectors, and the vast general public are searching for the best way forward to keep the major technology companies that shape our daily behavior accountable for the data they collect and sell. We know these practices have a manipulative effect, more profound than any other data capture/advertising business model of the past [1]. Developments in technology have allowed for a sophisticated Wizard of Oz-like presentation that hides how its algorithms interpret the data it collects and aggregates; however, concepts of transparency are thrown around as claims that the increasingly sophisticated technology can provide these qualities implicitly [2,3]. Some applications (i.e., Google, Facebook, Microsoft), for instance, make use of design capabilities to influence users' data privacy settings, a practice now being deemed "dark patterns" [1]. Yet, the 'smart' software embedded in many of our daily practices such as with pervasive computing and the internet of things (IoT) brings with it new opportunities for hidden contracts, automated data collection, and increasingly context algorithms. In response, new technological infrastructures such as blockchain applications explicitly tout anonymity, making use of decentralized computing systems that render tracing users nearly impossible. This has many benefits including encrypted records that are reportedly immutable, allowing for them to act as currency, notaries, authentication markers, even contracts [4], with the downside being that the anonymity offered can attract bad actors [5]. As blockchain proponents tout the transparency and accountability they say is built into its architecture, it is important that it is not considered a *solution* to the aforementioned issues without a rigorous critique of its methods for solving them.

Since its recent spike in use and popularity around 2016–2017, blockchain technology's uses have included cryptocurrencies, smart contracts, and non-fungible tokens (NFTs),

increasingly among others. Broadly, as a blockchain creates a ledger of immutable records, it can be diced up into portions that can be given value as currency (e.g., Bitcoin), can be coded to execute automated transactions (e.g., Ethereum), and can be tied to assets as a record of ownership (e.g., non-fungible tokens or "NFTs"). Past work in this space has discussed the potential dangers of assuming some of these blockchain technologies can replace certain types of documents that have standards and practices which have been around for centuries. For instance, the issues with standard form contracts can be exacerbated in this blockchain environment [6,7], the freedom of contract principle can be exploited to allow for the cementing of unfair terms [8], and, claiming that blockchain produces immutable records when the technology has facilitated nefarious activity can cause all sorts of issues for regulators—especially when these records are used for evidence [5]. Other research has noted the inconsistencies with the technology and some of its claims [5,6,9]. A sample of this research suggests that to solve some of these transparency issues the roles of responsibility should be considered; should blockchain applications have fiduciary responsibilities, for example? What are the proper notation practices as NFTs are essentially documentation of ownership? [9]. And who is responsible for solving its environmental impact? There are still many unanswered questions that need to be addressed to figure out the possible issues with this new digital environment.

Of the examples of blockchain technology mentioned, NFTs have gained renown for their large purchase price, bringing in sometimes millions of dollars to own one. Recently an NFT was auctioned off at Christie's for $69 million that a represented digital artwork, for instance [10]. At the time of writing (Aug 2021), the total trading value of NFTs in the last 24 h is $3,656,194,242.58 [11]. NFTs are generally created with the ERC-721 standard written in the Solidity language and make use of public blockchain protocols and platforms such as Ethereum, EOS, Cardano, Flow, and Tron, among others [10–13]. They are called "non-fungible" since they do not exist in a one-to-one ratio with other assets (such as cryptocurrencies do). Essentially, NFTs make ownership of an asset verifiable and since each asset is unique, each NFT record has a unique value relevant to the value of the asset itself. While NFT applications make use of other cryptocurrencies and are tied to their markets in some respects [13], they are not currencies themselves, but rather more like records of ownership. The IoT environment is increasingly making use of decentralized computing systems and blockchain applications to tie real world items to digital records that can prove ownership and trading rights. Sales of assets, transfers of ownership, even rental or insurance agreements might have a NFT associated with it. However, these applications will need to make use of automated smart contracts that control the terms associated with the transaction, bringing about a host of issues that go back to how users have engaged with standard form contracts for decades [7]. Additionally, industry has yet to reckon with the compliance issues presented with an anonymous technology, and, along those lines, regulators will need a method that will aid in updating outdated regulatory schema so that it can address these issues.

This paper adds to past research on the critical application of document standards to blockchain applications, in this instance, a study of NFTs as records. By 'document standards' it is referring to the types of mechanisms that have been applied to records and documents over time to maintain their authenticity. This draws upon contract law, document theory, diplomatics, records management, evidence and the descriptions for compliance as outlined in regulatory schema. These standards have been developed in nuanced, iterative ways with details that describe markers which provide actual validity and thus viability with documents. This could include chain of custody, records life cycles, even interrogating the epistemological underpinnings of how documents function and how users engage with them. As a methodology, it is not only theoretical and qualitative, but also pragmatic since there are real issues that need to be solved in a regulatory sense. Moreover, if these blockchain records are not interrogated sufficiently, the consequences would be that they *exacerbate* the issues with records and their possible nefarious uses, so it is imperative that these studies take place now. And although blockchain technology

is proving to contribute to environmental issues [14], when this aspect becomes more sustainable, it seems like this technology is here to stay and could transform business practices and records management across the space entirely.

While the conversation is being had around transparency for regular technology companies, blockchain technology has sat comfortably within this sphere due to the public nature of its records, the immutability of the information, and the supposed ease of which it is queried [10]. However, this paper argues that the transparency provided by blockchain technology largely serves either as a checkmark for firms or, at most, an easier information dump to sift through for regulators. The transparency that it provides does not necessarily translate to accountability. In fact, too much information can have the opposite effect if not curated properly [2,3].

This study examines current blockchain NFT implementations to test to assess their output to see if there is enough information to produce a viable record. NFTs were chosen since they blur the line between financial token and record of proof [9]. As the records produced from these applications act as both representative of value but also as something attached to another thing (being a digital file or real-world asset), the needs associated with the record are unique and worth rigorous analysis that could be generalized across the field. This space still needs much more work in terms of exploring the benefits of document standards for blockchain technology as a way of interrogating its actual levels of transparency. It should be made clear that this study is *not* conclusive in scope—rather, it simply examines the idea that the use of these document standards as points of measurement for various audiences or perspectives (i.e., Users, Firms, Regulators) could be helpful, and details a methodology to this purpose. It also calls for more research in these areas by highlighting where the work needs to be done. The results of this preliminary study show document standards are indeed necessary for this technology to grow and thrive in the way that is fair and transparent and in line with its motivation.

## 2. Materials and Methods

This section outlines the methods and materials used in this study of NFTs and provides a way of looking at blockchain records more generally. It builds upon the body of past research on the critical application of document and recordkeeping standards to blockchain applications, as well as appropriate document and records theories that have not yet been applied. This section defines the phrase 'document standards,' or the types of mechanisms that have been employed over centuries to records and documents to maintain their authenticity. This proposed method draws upon these fields that deal with documents, records, interfaces, and evidence to determine standards of measurement, rather than take for granted that an encrypted technological architecture will produce it automatically. These standards have been developed in nuanced, iterative ways over time and promote particular features that designate qualities such as validity, reliability, and authenticity to records and other types of documents [15,16].

This rest of this section details the terms that are operationalized and provides brief summaries of which document standards affect which stakeholder. It is intentionally hefty as a large part of this project is to provide a novel methodology, in addition to the case study, that can be used to study other NFT or blockchain instances in the future.

### 2.1. Terms Defined

This section explains the terms and concepts used throughout this study. Some of these terms are more self-evident than others. For instance, "blockchain" and "non-fungible tokens" are described in terms that other scholars and industry people have described several times [10,12,17]. Others, such as "document standards" are laid out in terms that have been aggregated from prior research. It is useful to define these terms so that this method is as apparent and replicable as possible. It is imperative that this space continues to get interrogated so that the research can help create the best possible uses for each instantiation of this technology.

### 2.1.1. "Blockchain"

A blockchain is a secure, distributed ledger comprised of a chain of record-type entities. These entities are cryptographically chained together and publicly replicated across each node of a decentralized computer network. There are generally two types of blockchain technology infrastructures—public and private ('permissioned'). Public blockchain infrastructure uses the transparency of multiple public copies of the ledger to ensure the accountability and accuracy of the entities. Since these are publicly copied onto each node of the peer-to-peer network, there is no centralized point of attack [18]. To fraudulently change records on the publicly distributed ledger, an attacker must control the majority of nodes in the network, a computationally expensive and improbable event. Some blockchain technology supporters promote variations of this technology such as smart contracts that aim to overturn centralized governance provided by third-party oversight with "immutable, unstoppable, and irrefutable computer code" that instantiates the "tamper-proof" records [19], which allows these 'contracts' the ability to "self-enforce" [20]. For a permissioned blockchain, the records are distributed within a closed network that requires permission; however, it still sports the features of consensus amongst multiple computing devices. Examples of public blockchains include Bitcoin, Ethereum, EOS, Cardano, Flow, and Tron.

### 2.1.2. "Non-Fungible Tokens" (NFTs)

An NFT is a unique cryptographic record linked to an asset, typically a piece of art, music, collectable, or another presumed valuable object [10,12]. These could be thought of as similar to trading cards for the digital age. Basically, NFT protocols provide an underlying distributed ledger for records, and combine it with transactions that make them exchangeable in a peer-to-peer network [12]. These records are called 'tokens' that can be bought, traded, or sold much like physical assets in some ways and radically different in others. Since they run on blockchain technology that claims to prove validity of the ownership of an asset, ideally all transactions associated with this relationship (i.e., between record and actual object) are recorded. NFTs are generally bought, sold, and traded from 'wallets' and can be explored, if public, on websites such as Blockchain.com, TokenView, and BTC.com.

There are a few notable features of NFTs that have been suggested to promote stability and consistency [12]. Figure 1 is a helpful diagram that shows the NFT process and the roles of each actor, reproduced from one of the first systemized studies of NFTs [12].

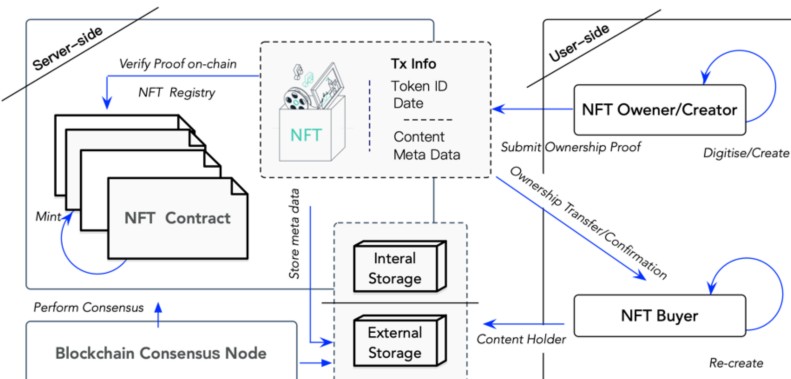

**Figure 1.** "Model of NFT Systems" (taken from Wang, Qin, et al. "Non-fungible token (NFT): Overview, evaluation, opportunities and challenges.").

Essentially this process consists of two roles: NFT owner and NFT buyer. An NFT owner digitizes the raw data from the transaction into the proper format, then stores it on a database external to the blockchain (or on a blockchain, but that is more costly). The owner then signs the transaction with a hash (or cryptographic signature made of a string of numbers) and sends the data to a smart contract. The smart contract processes the data, then

mint or trades it on the blockchain as a transaction. Once it has been confirmed through mathematical consensus, the NFT is linked permanently to the unique hash identifier and the distributed blockchain record, where it cannot be changed.

### 2.1.3. "Document Standards"

This paper utilizes the phrase 'document standards,' which stands in for the multiple, interdisciplinary mechanisms that have been applied to records and documents over the centuries in order to maintain their authenticity. This draws upon document theory [21,22], diplomatics [12,13], interface theory [23,24], evidence [25–27], and regulatory schema that depend on records management, thus speaking to the interdisciplinarity of this method. These standards have been developed in nuanced, iterative ways over time with details that describe markers that provide actual validity and thus viability with documents and records. This includes how chain of custody and ownership is maintained, which information on users should be stored, even interrogating the epistemological underpinnings of how documents function and how users engage with them. This body of work was compiled organically from various fields, but they all share a common humanistic or sociological attention to the subject or user position that allows for a type of activism throughout their arguments. As a methodology, it is not only theoretical and qualitative, but also pragmatic since there are real issues that need to be solved in a practical and regulatory sense.

### 2.1.4. "Users"

This project separates off each of the stakeholders into general categories. The first of these categories is 'Users,' which includes any person that makes use of blockchain technology in any capacity. This may include people who are using a public blockchain administerd in a decentralized way, and also those who administer the technology in permissioned chains for their own purposes. It does not include the technology itself, for which there is a good argument to be made that there is a certain amount of agency that can be coded into the technology (for example, as a 'user' who perpetuates certain transactions on the blockchain such as smart contracts). Rather, this category is purporsely excluding technology from this category and looks to any real person whom did the coding at some point. This helps determine where the responsibilty lies when studying accountability and transparency issues in practice.

### 2.1.5. "Firms"

The category called 'Firms' is defined as any business entity that adminsters, facilitates, or incorporates a blockchain into its infrastructure. This includes public blockchain offerings that create the design for tokens which are 'mined' for consensus with the computing power across peer-to-peer devices. It also includes the permissioned blockchain adminsters whom incorpoarate blockchain databases into their current business environment. Lastly, it includes the intermediary companies that provide the services which facilitate blockchain applications; for instance, those that offer 'wallets' to allow for the buying, selling, and holding of tokens, or 'explorers' that allow for the reading of the cryptographic records of public systems. Examples of 'firms' include Ethereum (which provides the protocol and public blockchain), eToro (which is a platform that makes use of Ethereum protocols to create NFTs), and even Christie's Auction house (which is now selling NFTs).

### 2.1.6. "Regulators"

The third study looks at blockchain technology from the perspective of 'Regulators,' which includes any entity whose job it is to produce policies and practices that would protect those in the category 'users' from any malfeasance that originated in the 'firms listed in the second category. These institutions have the vast task of sorting through a technology that may have already outpaced current policy. This is especially important since some of the orignal motivations behind blockchain technology was to sidestep these

exact regulatory bodies [19]. Current regulatory schemas are being developed for big tech companies across the board, including the General Data Protection Regulation (GDPR) by the EU and the California Consumer Privacy Act (CCPA) in the US, which push for more transparent disclosure practices and protections for patrons of these applications. In the financial space, regulatory agencies have already begun to tackle blockchain applications (e.g., Financial Crimes Enforcement Network [FinCen], Office of Foreign Assets Control [OFAC], Internal Revenue Service [IRS]). This study tries to be aware of the financial regu**a**ltory space as it is highly specialized and developing, yet **stll** applicable to NFTs in a similar fashion as cryptocurrencies and smart contracts. Moreover, it draws upon the new consumer protection regimes (e.g., GDPR and CCPA) to consider that other types of regulatory schema that no doubt will be applied to blockchain applications as well.

### 2.2. Project Design

This project is designed so that it provides a methodology for examining certain qualities of blockchain records that would provide transparency and accountability. It takes the definitions in the previous subsection and applies the category 'document standards' to 'blockchain technology' (specifically NFTs) to see if they are up to par and to imagine how they might be improved. It does so from the three perspectives of (1) Users, (2) Firms, and (3) Regulators. The following details the queries that were conducted for each perspective. The results of these queries are in the next section.

#### 2.2.1. Queries for Users

- Copyright: How do you clearly and conspicuously "attach" terms and conditions to an NFT and ensure that those terms follow the NFT and bind subsequent owners? How are evolving copyright issues handled (public domain laws, estate changes, etc.)?
- Standard form contracts: Do these tokens present the same issues as other standard form contracts if this same genre of contract is used to lay out terms?
- Interface design: Do these records match user expectations and follow usability and design best practices? Are they designed with users in mind? Who is their audience?

#### 2.2.2. Queries for Firms

- Jurisdiction: If a Firm makes use of a public blockchain to create their NFTs, do they streamline transactions or obscure necessary data (e.g., accd. to FCPA, SOX, UETA)?
- Liability: Do the current records protect from liability or increase risk?
- Compliance: Do the current NFT records aid or hinder other compliance or record-keeping efforts?

#### 2.2.3. Queries for Regulators

- Investigation: Do the records produced by NFTs allow for the same procedures when investigating a bad actor?
- Evidence: Do the current records serve as viable evidence? What are the consequences if they do (accd. to the US Federal rules of Evidence) but do not have the required components of legitimate records (accd. to document standards)?
- Policy: How does blockchain aid or hinder the regulations set forth by KYC, OFAC, FASB, SOX, and GDPR, among others?

### 2.3. Limitations

Although effort was put into ensuring that this study is as sound and replicable as possible, it does have its limitations. This methodology follows the dominant research method from the past several decades, "interdisciplinarity," which has benefits in its application to the perspectives of multiple involved parties or stakeholders. The limits to this method, however, are that it has been accused of overlooking potentially contrasting assumptions in each discipline, promoting what was identified early on as "conceptual confusion" [28]. This often occurs through the motivation to find similarities between

the disciplines that may not exist; but it can be remedied by noting explicit differences and contexts throughout the analysis, including between vastly different origins, history, objects of study, and interest [29]. While there is not enough space in this paper to give each of these fields the full treatment they deserve, the interdisciplinarity of this study is still useful as the fields utilized (that make up the category called "document standards") work toward the common goal of maintaining the qualities that determine whether a record is viable.

Additionally, some of the advantages of interdisciplinary research that this method hopes to benefit from include some logistical advantages such as a wider audience and other, loftier goals such as possibly more 'normative' conclusions, which might be more well-rounded and humane and consider "trade-offs" and "principles" (including ethical concerns) [30]. Also, writing for a larger, multi-disciplinary audience not only encourages more productive solutions that may not have occurred otherwise, but also provides a close textual study that is qualitative and holistic.

## 3. Results

This section describes the results of exploring the above questions, including the interpretation of the results and what it might mean for blockchain technology going forward. Below, the results of the application of document standards to the aforementioned queries from each of the three perspectives (i.e., Users, Firms, Regulators) is shown.

### 3.1. Description of Results

This section will describe in clear, succinct summaries how these queries were considered in this study. The document standards that were applied to each query are explained, as well as a brief acknowledgment of the context within which these standards were developed. The purpose of these results is not to prove that these queries *have* been answered by this study; rather, it is to show that these questions exist and that they *can* be answered.

#### 3.1.1. Results for Users

In the case of NFTs, their purpose is to tie a record to prove the ownership of an asset, whether it be a digital file or a real-world item. While at first this may seem promising because it is a more immutable record due to its cryptographic qualities, this record can present very significant issues. First, most records of ownership that are used for this purpose generally need to have several qualities for them to be viable, including reliability, validity, and authenticity [12,13]. These qualities come from certain information being documented, demonstrating chain of custody and ownership, and most importantly to answer the queries for Users, the terms of this ownership over time. This last concern requires that the record makes use of a contract that will lay out the terms of copyright, use, transfer of ownership, etc. The concern here centers around two issues; (1) that the NFT record will begin to use *standard form contracts* for this purpose, which includes all the issues associated with this genre of contract [31–35], (2) that the immutability of the record will enforce these terms beyond even what normative contracts negotiate, not allowing for the flexibility that contracts require, and (3) that the design of the NFT record will further obscure any important information for the User with unfamiliar or deceptive design practices. These queries are answered within the history and context of contracts and design since they are a vital part of these records being viable for asset ownership.

As warned about in the literature around standard form contracts (e.g., Terms of Service), these 'zombie contracts' may appear as traditional contracts on the surface, but under deeper scrutiny, have "several distinct features that sit in very deep tension with contract [doctrine]" [31]. Contract law is generally viewed as a remedial "institution" whose function is to adjudicate any issues that arise between two individuals or entities after transactional activity as they arise [32]. Generally, contract law enables two or more self-governing parties to document shared goals. With *standard form* contracts, however, one party drafts the terms, relying on past types of similar contracts. In other words,

notions of 'standardization' here are being defined and perpetuated by 'standard practice.' Without regulatory response to inhibit egregious terms, one party of these contracts is at a vast advantage in the arrangement and, further, there are concerns that powerful players could take control of their governance, ensuring any regulations result in their favor. [33].

Recent studies confirm that standard form contracts reinforce certain inequities between classes—not only through unregulated clauses, but also in terms of who advocates for fairness. Because of a difference of perception and knowledge of contracts amongst users, which varies according to socio-economic class, "elite customers" are less likely to advocate for the fairness of the agreement and are only motivated to stop egregiousness when it threatens them personally, further solidifying the distance between the "haves" and "have nots" [31,34]. This both reinforces a general distrust in the legal system for the lower classes as well as reactionary costs for the Firm such as evasion measures (e.g., piracy, hacking, misuse of services) that might come about consequently. Since there is a great amount of anonymity with blockchain, contracts used in an NFT situation where owners and traders/buyers are not identified could exacerbate some of these power imbalances with more immutable, boilerplate terms—an even stronger win for the more powerful entity with no path toward restitution other than those that are illegal and put other users at risk.

However, there are some areas where blockchain technology could help with securing contract terms. One of these areas is with unilateral modification clauses. These common clauses allow for a service provider to change the other terms of the contract at will, essentially making the other promises in the contract "completely illusory" and at the determination of only one party [35]. In other words, for a user, the concept of a contract document as a stable entity is disrupted by the mere fact that it could change at any time without their knowledge of this change. As Preston and McCann (2011) ask: "If the service provider can change the contract at will, why bother to call it a contract at all?" [35] If the records provide more concrete documentation, it could potentially provide a chain of records that prevents this clause from being enforced, which is significantly better than the Terms of Service agreements that change at will without a record of the past terms. There are normative benefits to contracts that have the flexibility to change terms as the relationship between the parties develop, and blockchain-made smart contracts have been critiqued from this standpoint prior [6]. These issues are not unfixable if thought is put into them before the issues that are already present are cemented into the genre. There is a body of research that is dedicated to studying the regulation of the issues associated with these contracts and it should be considered if they become a part of the purpose and practice of the records that support NFTs.

An additional area of study that could help NFT records from the perspective of Users, especially since the result is user-facing, would be the area of interface and document theory. These disciplines work toward understanding the concepts of documents and interfaces, both of which would be useful in the process of understand what users expect from these records. If they remain somewhat exclusive and specific to the realm of blockchain output, the average user will not find them meaningful, and consequently not be as transparent as they purport to be. For instance, information and media theorists have expanded on earlier ideas but locates the 'informativeness' of a document (and thus and definition of information itself) in its materiality, institutional embeddedness, and historical contingency, and recognition of user subjectivity, rather than in a theory that assumes an 'intentional substance' of a phenomenon [21–24]. In other words, once all the social and political forces that configure documentary practices are considered, "the genie is out of the bottle: the informativeness of documents, when recognized as dependent on practices is also dependent on what shapes and configures them" [21]. This body of work provides an analysis of documentary agency and seeks to understand how even fundamental understandings of a document, text, or interface can be undermined by a design that is inaccessible and unfamiliar [23,24]. This way of thinking is not only beneficial for Users,

but for all stakeholders as it considers the audience and context of the record, thereby making the outcome more useable, more transparent, and, ultimately, more accountable.

3.1.2. Results for Firms

The biggest concern for Firms in regards to blockhain technology is the issues that come about from compliance policy and regulation. Some US laws that affect blockchain applications include recordkeeping determinations of the Foreign Corrupt Practices Act of 1977 (FCPA), the Sarbanes-Oxely Act (SOX), and the Uniform Electronic Transaction Act (UETA) of 2000. Civil systems such as the E.U. have also sought to find ways to keep coroporations accountable with the information they store and disclose, including GDPR (and subsequently California's CCPA), for which eventually Firms from the blockchain space will also be held accountable. As a new sphere of corporate activity has surrounded blockchain technology, these new companies, which are often start-ups, are concerned with liability and compliance as they coud become major issues to their endeavors.

Produced in reaction to a string of several major corporate scandals such as Enron and WorldCom, SOX was the most comprehensive accounting reform enforcement since the FCPA from the late 1970s [36]. Both regulatory schema outlined accounting and bookkeeping requirements for companies, with FCPA handling transparency in dealing with foreign officials and SOX concentrating on the alteration and destruction of records. Both acts also endorse the need for more thourough recordkeeping requirements to prevent and prosecute white collar crime and fraud. Specifically, the FCPA requires companies to make accurate and complete records and devise methods for an internal system of accounting controls [31]. SOX expanded on these basic requests, asking for transparent and accurate corporate records (Title III, Section 302) and both internal and external auditing assessments. It also outlines specific actions regarding the destruction of or tampering with records involved in "official proceedings" (Title VIII also called the "Corporate Fraud Accountability Act of 2002"). Although blockchain records claim to be immutable, there are ways to code in acts of deletion with smart contracts or to fork the chain, which changes the outcome of the order of the records. If this happened with a public blockchain application, a bad actor could code in the deletion of the record, violating SOX and making the Firm liable for sanctions and fines.

On the other hand, the UETA laws of the early 2000s were meant to streamline commercial activity online across multiple jurisdictions [37], yet in doing so, lowered the recordkeeping requirements for digital transactions, including removing the retention of paper copies and relaxing the types of actions that signal agreement amongst parties. These laws ultimately benefit NFTs and blockchain technology as they lower the standard for compliance and do not require any extra disclosures on the part of whomever is managing the blockchain (the viewers and wallet companies, for instance). This only goes so far, however, as other more recent laws supercede these and require these types of companies to retain knowledge of their customers for money-laundering reasons (remember, this does not the blokchain itself, as no one is responsible exactly for those records).

Newer EU policies, including the General Data Protection Regulation (GDPR) that was enacted in May 2018, specifies, among other principles, an accountability mechanism referred to as "the right to know" that describes corporate responsibility to inform consumers about any collected personal data and the algorithms that affect a user's experience in this regard [38]. Clarity of information to the public is one of the major principles of GDPR and requires transparency in terms of the contracts that govern the way this information is handled. From these nerw regulations came other regulatory schema have been formed such as the California Consumer Protection Act (CCPA) that requires transparency and disclosuresfor technology companies and complicates this already complex regulatory landscape. It is difficult to know how these new regulatory paradigms will play out with blockchain technology since NFTs are built publically and are used for many different purposes and for various 'audiences.' And while the one of the limitations of this study is the simplification of these groups of interested parties, the study overall is useful in its

ability to consider the liability that the firms whom facilitate these applications might be subjected to.

### 3.1.3. Results for Regulators

Regulatory bodies are concerned with how they might protect consumers and the public from corporate activity they deem harmful. Generally, this process of accountability involves either preventative measures such as compliance requirements or accountability measures after the fact that investigate and collect evidence for prosecution. With blockchain technology, this process can be hindered by the issues with authentic documents and how these documents serve as proof for investigative purposes. As mentioned in the last section, preventive measures and compliance do not always lead to easier culling for regulatory bodies or prosecuting entities. This section focuses on their perspective and the difficulties blockchain technology presents when building a case against a Firm, which looks to the rules and uses of evidence to determine how effective these records are in this realm.

Records have long been associated with manifestations of evidence, even if evidence can take many other forms and instantiations [26,27]. The history and uses of evidence suggest that understandings of what comprises evidence are dependent upon the paradigm within which evidence will be acquired, assessed, and introduced, not just simply how they support an argument logically. For example, even if a record is produced that seemingly claims to prove something over something else objectively, the record itself must prove that it is an authentic document before it can be used in the case. Thus, the circumstances around the record and its creation, history, uses, etc. must also be rigorously considered. This project makes the case that considering legal conceptions of evidence from a more holistic, philosophical perspective such as in the research of legal scholar John Henry Wigmore (1863–1943), is a more useful route than simply relying on static definitions such as those laid out in legal doctrine.

Wigmore's work argued that "facts are evidence insofar as they play a role in a teleologically directed argument" [25]. In other words, his work found that the process of using evidence to prove a case required viewing it as three distinct layers of information: (1) as a proposition (hypothesis); (2) as specific elements of law that need to be satisfied; and (3) as material evidence and facts that make up the narrative of the case. This view of evidence fixes the "worn out legal system" of the nineteenth century that relied mostly on numerical systems and had "no understanding of the living process of belief." This paradigm, while over a century old, is useful to consider as it brings back the nature of human judgement and essentially what is overlooked if blockchain technology stands in for legitimate records.

Wigmore distinguishes the uses of evidence into two categories: (1) the analysis that details the informal logic of reasoning and argumentation, which he called Proof, and (2) the "rules of procedure," which he called Admissibility. Proof consists of the practice concerned with "the ratiocinative process of continuous persuasion." Admissibility consists of the procedural rules developed by the law and based on "litigious experience and tradition, to guard the [jury] against erroneous persuasion." These distinctions can help us consider the "probative force" of a piece of evidence, which describes its tendency to support or to negate the first piece of information, the proposition or hypothesis. A record in this paradigm could have two outcomes once it is interpreted in a trial: Proof or non-Proof. Currently, the Federal Rules of Evidence govern the process of evidence admissibility and discretion in trial court. A difference in Wigmore's conception of evidence is that he makes distinctions between other aspects Proof, including argumentation, inference, and probative value, rather than superficially codifying them into procedure doctrines [25]. This project confirms the usefulness of these early distinctions by Wigmore, providing a rigorous method of logical inquiry that benefits current evidentiary paradigms in which these aspects might be discounted.

A more holistic and humanistic understanding of evidence is especially important since blockchain records could give the false sense that somehow, they implicitly have the qualities needed for Proof—the issue being that unless the technology and infrastructure of the blockchain being considered is scrutinized properly and understood thoroughly, these qualities are not ensured. In other words, if an investigator or regulator or even a jury or judge misunderstand the technology behind a blockchain record and think that somehow it purports some type of reliable information because of its cryptographic features or architectural design, these records will be *more* dangerous than had they not been made of blockchain. The technology could provide a misleading testament to validity when actually they could *aid* in nefarious activity.

### 3.2. Interpretation of Results

The results (Table 1) can mean only one thing for blockchain environments—more work is needed before they accomplish the transparency it claims to provide. Below a screenshotted is provided of two recent NFT records taken from the blockchain explorer Adapools.com. These records show the information that is typically displayed on a blockchain record. It is also useful to see a real example of a NFT record to analyze how it might be improved to address the issues just mentioned. Newly proposed methods [39] that take apart the components of a document could consider the components as they currently exist and reconstruct the design of the record to include appropriate context and information from each of the three perspectives. The rest of this section details how a few important aspects of this example NFT record could be improved.

**Table 1.** Description of Results.

| Perspective/Stakeholder | Area of Concern | Applicable Document Standards | Examples |
|---|---|---|---|
| *User* | • Design<br>• Usability<br>• Ownership | • UX/Human Factors Principles<br>• Contract Literature (esp. standard-form)<br>• Copyright Literature | • Clear coherent design that considers the needs of user ownership over time, including:<br>(1) "Name"<br>(2) "Publisher"<br>(3) "Collection"<br>(4) "Artist"<br><br>• Fair contract terms<br>• Appropriate contextual elements that utilize familiar design conventions<br>• No deceptive design that perpetuates inequities |
| *Firm* | • Assets<br>• Recordkeeping<br>• Compliance<br>• Risk Assessment and Reduction | • Recordkeeping standards (i.e., validity/reliability)<br>• Asset document conventions and standards<br>• Electronic contract standards and compliance (e.g., UETA, FCPA, AML, SOX)<br>• Transparency requirements such as data disclosures (e.g., GDPR, CCPA) | • Storing customer information for AML and preventing deletion of records for SOX<br>• Current asset record requirements:<br>(1) Description<br>(2) Location<br>(3) Procurement<br>(4) Life Cycle<br>(5) History<br>(6) Depreciation value<br>(7) Insurance<br>(8) Maintenance<br>(9) Ownership distinctions<br>(10) Barcode or serial number<br>(11) Warranty information |

**Table 1.** *Cont.*

| Perspective/Stakeholder | Area of Concern | Applicable Document Standards | Examples |
|---|---|---|---|
| *Regulator* | • Evidence<br>• Protection<br>• Transparency | • Evidentiary requirements<br>• Discrete regulatory schema that are effective | • Viable evidentiary components, including:<br>(1) Parallel contracts in other mediums<br>(2) Searches, subpoenas<br>(3) Payment history between parties<br>(4) Communication between parties (e.g., phone calls, email)<br>(5) Witness accounts<br>(6) Blockchain environment-specific conditions, (e.g., authentication protocols, automated features, triggers, oracles)<br>(7) Personal computer evidence (acquired through search warrants)<br>(8) Wallet managing services, along with ISPs, phone records |

For Users, the two most important considerations are attached contract terms and usability. From Figure 2a,b it is apparent that the information is neither contextualized nor understandable in terms of the normative conception of asset records of ownership. Further, although the cryptographic hash promises to make the record unique (which is does in a technological sense), it is difficult to decipher the difference between the digital objects when similar records are inquired about as you can see in Figure 2b. In terms of the contract issues, this record is not sophisticated enough to have attached terms; however, one could imagine what that might look like in this instance. Some of the initial, basic information for a contract arrangement is present, yet not clear—the two parties could be the "Name," "Collection," "Artist," or "Publisher." It is not apparent from the information shown what the relationship is between these entities. Moreover, if the basic ownership information is not clear, then the projected trajectory of their relationship is even more unclear. In a contract situation, this information would be clarified upfront in a Preamble, for instance. As an example of how much contracts could be skewed in this environment, consider the effort that generally goes into the conventions and allowances of very nuanced language choices in a non-blockchain produced contract [40].

Additionally, the anonymity of these entities allows for the record to exist publicly and be authenticated on a public blockchain. While this is important for 'transparency' in this respect, if a contract situation is entered, which in the case of a NFT it seems would be quite common, then it could preclude clear and conspicuous terms from also benefiting from this transparency. If there is a power imbalance (or information asymmetry) between the two contracting parties, this might be especially concerning [34].

For Firms, the current design of NFTs could confuse some of the requirements and responsibility of a corporate entity. As Firms are concerned with liability, an ambiguous set of metadata might not satisfy the information needed for FCPA. As the anonymity can obscure the location information, the rules for foreign transaction are also obscured. This might cause the company to be at risk for compliance issues unless this information has been obtained and stored, which would then reduce the anonymity that allows for the transparency that produces immutable records. This is one area that needs much more focused attention—sorting out personal information, privacy, and compliance while negotiating the needs of a decentralized architecture and maintain the integrity it claims to promote.

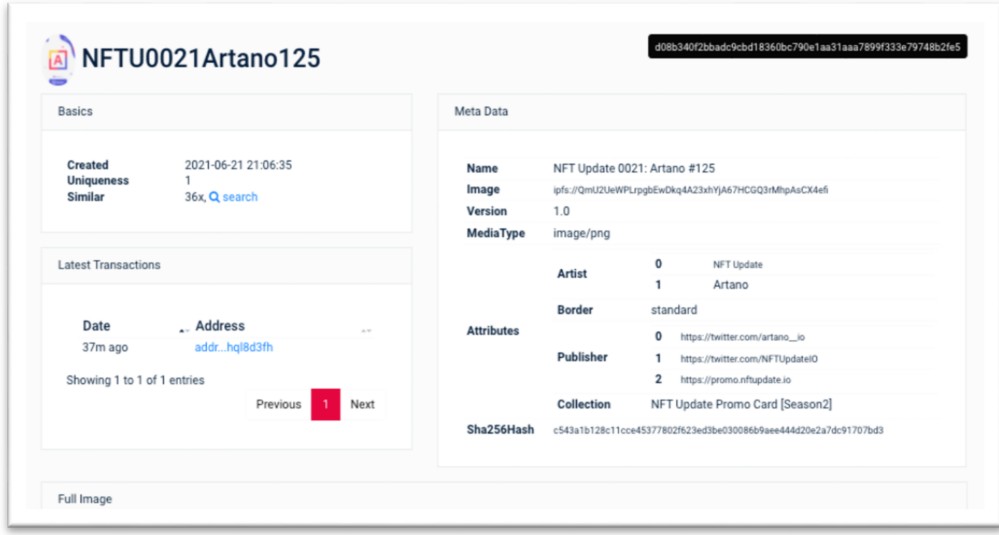

(**a**)

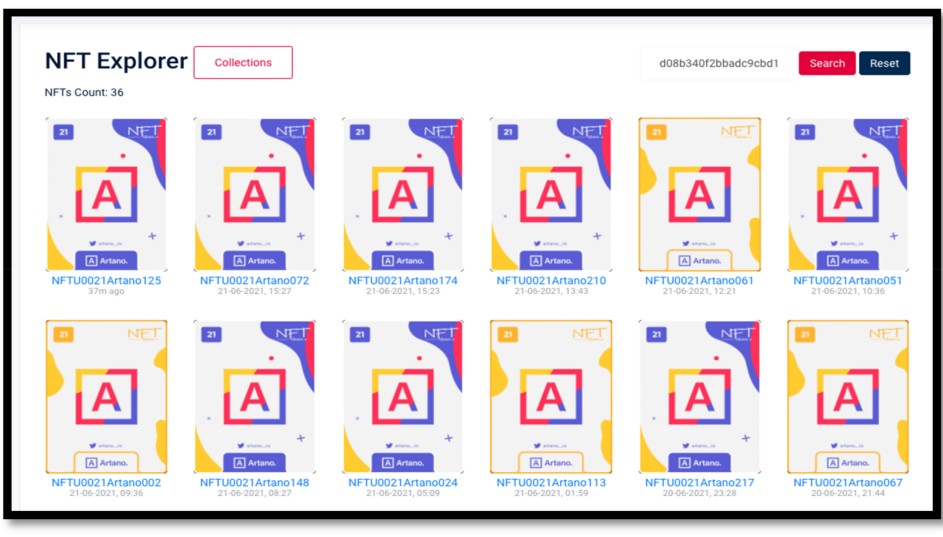

(**b**)

**Figure 2.** This is a figure. Schemes follow another format. If there are multiple panels, they should be listed as: (**a**) An example of a public NFT record; (**b**) A search from the "Similar" link on Figure 1. Source: Adapools.org.

Moreover, it could be argued possibly that blockchain records have simply been streamlined for easy transaction under the UETA laws. Yet, if nefarious activity such as money laundering took place—for instance, if someone 'bought' an asset with tainted currency and makes use of the anonymity that it provides, the transaction could be a liability for the company. Like foreign transactional requirements, if the company does not retain the information about the parties behind the NFT, the laws that require financial institutions to have this data such as Know-Your-Customer (KYC) laws. If a participant in a NFT transaction is sanctioned by the Office of Foreign Assets Control (OFAC), this could be a major violation of the rules set forth by FinCen or other agencies. Since NFTs are records of ownership, it makes sense to look at how these types of records have been considered previously. This includes how they are situated, criticized, used as evidence, retain their authenticity, and more. For example, is important to know what information has been required on these types of records. Then, when concerning NFTs, it should be negotiated how this information will be displayed or retained with privacy concerns. For instance,

blockchain technology in general might have trouble complying with new disclosure laws (i.e., GDPR and CCPA) and something like 'privacy by design' could be necessary. Making use of UX and interface design theories (as outlined in the Users' section) could also aid firms in this task [41].

Additionally, it is in the interest of a firm to correctly manage their assets so that they can accurately document the value of their company and avoid unnecessary risk. This is where considering previous document standards are useful. Asset records, to be useful, must contain certain information according to the financial and business world. This includes: (1) a description of the asset, (2) the exact location of the asset (what about link rot? is it hybrid of local/cloud etc.), (3) procurement details including purchase dates and price history, (4) life expectancy, (5) depreciation value, (6) insurance and compliance details, (7) maintenance history, including repairs and downtime, (8) the owner of the asset versus the user of the asset, (9) the barcode or serial number, (10) warranty information, and the list goes on. It benefits the business to have all this information since a lot of these transactions are *financial* and money laundering is a real concern. Anti-money laundering (AML) rules require that a company know their customer (KYC) and have certain data on them or else they become liable. In the video game world, it would be worth considering who would be responsible in this case—the video game creators or the group that maintains the NFT protocol? In the blockchain world, this needs to be negotiated with the movement that is motivated to be both transparent *and* anonymous. Obviously, including all that information on an NFT record would be a task that needs some thinking in terms of design.

For Regulators, the main concern in terms of blockchain technology is how it aids or hinders their goal of keeping Firms accountable and providing the proper scrutiny to stop the disreputable uses of its features. This includes the rigorous examination of blockchain records as evidence, including not accepting that the technology itself creates validity or reliability in a diplomatic sense [15,16]. As Wigmore points out, the creation of an argument using evidence such as those used in litigation, does not depend solely on the qualities of the records. These arguments are contextual and if all parties involved do not understand the records fully, which is nearly impossible with NFT records in their current form (Figure 2a), then the case will not be fully examined appropriately. This could have major consequences down the road if the precedent is set that the records *are* viable in their current form. The qualities they purport to have will never actually be developed. For instance, in previous work, a criminal investigator from the U.S. Internal Revenue Service (IRS) provided a list of all the information needed for prosecution of a blockchain case that involved smart contracts [7]. This list includes:

(1) A parallel paper contract with agreements from both parties (public records
(2) Searches, subpoenas to record keeping authorities)
(3) A history of payments between A and B (subpoenas to corresponding financial institutions)
(4) Email, Phone, other types of communication (subpoenas or search warrants)
(5) Undercover, if necessary (to reveal and confirm any allegations of fraud on the part of the various parties)
(6) Witness accounts from the inception of the agreement through the present (interviews)
(7) Conditions on the blockchain that would reveal payment/non-payment
(8) Their corresponding truth/falseness in real life
(9) Authentication protocols for both users and their conditions (are the transactions automated or triggered by one party or the other, and for what reason?)
(10) Personal computer evidence (acquired through search warrants or consent searches)
(11) Wallet managing services, along with ISPs, phone records and other third-party record keepers would be acquired through subpoena or court order.

The list is quite thorough, but it gives a sense of what could be required to build a case against a Firm. While it is not practical to assume or even hope that a Firm will be required to provide this information since one of the major features of blockchain application is the benefits of an anonymous system, if Regulators are serious about preventing or prosecuting bad actors, then some thought is due in this area. Perhaps there is a way to keep the

information in a secured location so that the Firm can comply with regulatory schema—however, this would require that it be kept in a centralized system, which ultimately would undermine the protection that a public chain offers, and which patrons currently enjoy.

## 4. Discussion

One of the most important aspects of the issues highlighted by this discussion is that the main feautre of a public blockchain—its transparency, which allows it to perpetuate claims of accountability—does not *ensure* that this accountability takes place without other considerations. While private information is generally protected, this anonymity also allows for the easier facilitation of nefarious activity such as contracts with unfair terms created by asymmetric power arrangements, ambigous and unfamiliar design, compliance issues, money-laundering issues, and more. Thinking through some of these issues premptively and prior to these practices becoming normalized could help begin a real, sustainable future for blockchain. Future research on this topic could help negotiate privacy concerns and anonymity benefits with transparency claims, for instance. Pinpointing responsible parties and maintenance issues, deletion concerns, and record metadata requirements is a daunting task with distributed infrastructure in some respects, but is also a worthwhile endeavor so as to make sure that the features of the technology are not abused.

There can be a few conclusions drawn from this preliminary study. The first is that the current implementations of blockchain technology need a bit of brainstorming and creative work in terms of its records and the audiences for which they are intended. Simplified conclusions from this study are as follows:

- *Users*: Blockchain records for Users could be developed with standard form contract issues in mind, as well as with usability and interface theory conceptualizations of how users engage with the technology and the records it produces. This might include designing the records so that they include context and familiarity, which would be useful in holding the parties of a contract accountable and providing those with the information they need to litigate a more powerful party. It could also normalize some of these design conventions so that the average person could understand their information, creating best practices that could iteratively be made better moving forward. *This is especially important for IoT applications as they are embedded into the fabric of everyday life, proliferating contracts for each instance of each transaction.*

- *Firms*: Blockchain records for Firms could be improved with some research that focuses on the information that should be required for companies to retain on their own records. Also, it should be considered how this would contribute to compliance or a lack thereof. Negotiating privacy concerns with the information that the company needs to be compliant is one of the biggest concerns for Firms at the moment. Reducing risk and liability for this new type of company encourages innovation in this space *as well as* sustainability. More users will begin to make use of blockchain applications as trust in the companies increases, which will help streamline some of the compliance requirements into more practical applications.

- *Regulators*: Blockchain records present several quagmires for Regulators. Since the decentralization of the blockchain ledger offers protection for Users, it is difficult to negotiate this protection with the necessary information needed to keep the Firms accountable. One possible way forward is to create personal data stores for consumers that could be accessed in an investigation [41]. There are still issues with this idea such as the inability to access this data due to multiple entities being involved, yet it could be studied as a model to prompt the sorting out of what information needs to be on the blockchain and who is responsible for its retention. Making nuanced and effective regulatory schemes is a concern being tackled for all technology applications. Along these lines, new disclosure laws such as GDPR and CCPA can be satisfied as effort is put forth in this space.

Ultimately the main conclusion that can be drawn from this study, and why it is named 'Betraying Blockchain ... ,' reiterates that although it seems like the technology

associated with these records proves a sense of viability in its immutability or transparency, those qualities should not be taken for granted as there is still much work to do in this space. And while blockchain, smart contracts, and NFTs are seemingly solving some of the issues present in the currently popular centralized data-collection application model, they cannot solve them adequately unless the context and situation of the application is considered. That includes looking at how this new technology replaces the past functions of records and how well it lives up to solving the problems associated with that practice. The IoT future will include applications that are involved in many aspects of our lives and involve sensitive data, so simply switching to a different type of infrastructure should not be thought of as a cure-all. Hopefully this work will free up interested parties to make use of more interdisciplinary research that moves from just cryptography to document and record theory and practices. The technology is exciting, but the future that awaits is dependent on us making use of and building upon the work that has come before us. This will then, in turn, support the goals of those utilizing the qualities that the technology should provide as the records can live up to the standards that have been developed over many centuries; only then can blockchain technology can be considered a 'development' in its truest sense.

**Funding:** This research received no external funding.

**Institutional Review Board Statement:** Not applicable.

**Informed Consent Statement:** Not applicable.

**Data Availability Statement:** Not applicable.

**Conflicts of Interest:** The authors declare no conflict of interest.

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
