# Peer review of "Betraying Blockchain: Accountability, Transparency and Document Standards for Non-Fungible Tokens (NFTs)"

_information, doi:10.3390/info12090358_

Round 1

Reviewer 1 Report

In this article, the main topic focuses on blockchain technology, analyzing its transparency offered by the infrastructure and record keeping mechanisms. Thus in the paper is presented a comparison between records produced by non-fungible tokens (NFTs) made using the blockchain and compares them with 'document standards', to see if they live up to the bar that has been set by a body of literature concerned with authentic documents. Following the analyzes performed on the transparency of record keeping, it was shown that without a design effort these recordings can only be an illusion and could serve the opposite purpose for bad actors. The overall conclusion from the research was that although records using this technology provide a sense of viability in its immutability or transparency, further work is needed in this area and these features should not be taken for granted.

The information in this paper is well presented and well structured. The theory presented is supported by comparisons and analyzes made by researchers. Abstract and conclusion are strongly related to the content of the article, and both correlate with each other.

However, the overall approach of the project should be presented graphically.  The suggestion for improving the quality of the information is to update the old references and very important, more figures, diagrams and graphs must be used to support the research results.

The main focus of this article is on Blockchain technology. Within this paper, a study is performed that analyzes the transparency of blockchain networks offered by the infrastructure and the mechanisms for keeping records. The comparison between the records produced by non-fungible tokens (NFT) and the records in the standard documents wants to show that detailed research of this technology is still necessary, even if, at first sight, it could be said that the network is immutable but also transparent.

More references should be added regarding different distributed ledger technologies, for example:

- Wang, Qin, et al. "Non-fungible token (NFT): Overview, evaluation, opportunities and challenges." arXiv preprint arXiv:2105.07447 (2021).

- Nadrag, Carmen, et al. "Comparative analysis of distributed ledger technologies." 2018 Global Wireless Summit (GWS). IEEE, 2018.

- Ante, Lennart. "The non-fungible token (NFT) market and its relationship with Bitcoin and Ethereum." Available at SSRN 3861106 (2021).

Author Response

First, thank you for your comments. You were generous in your assessment of the conclusions of my research and the clarity of its presentation and structure. I am pleased that the point of my paper was communicated, and your suggestions went a long way to make that even more so.

To your individual concerns:

-You asked me to present my conclusions more graphically. This was a helpful comment. I inserted a table that bullet points the document standards and conclusions, as well as separates them by each stakeholder’s perspective. It is an extensive chart that I believe really serves to compliment the analysis and to highlight the specific examples that correlate with each audience. I also made sure that the formatting of the rest of the document, including the sample NFT, is correct.

-You asked me to update my references. I added in the references you suggested (which were great by the way and should have been present prior to your review). I also updated other sources with more recent research.

Thank you again. I hope my revisions satisfy your concerns.

Reviewer 2 Report

This paper analyzes and compares records produced by non-fungible tokens with 'document standards' through an interdisciplinary method. 

The Introduction presents the relevance of the problem. The authors argue that the transparency provided by blockchain tech it gives does not necessarily translate to accountability. The paper shows that the benefits of document standards for blockchain technology may improve the actual levels of transparency.

The second section describes the methodology, details the terms used, and summarizes which document standards affect which stakeholder.
Yet, in section 2, the authors present their methodology for examining transparency and accountability from the three perspectives of users, firms, and regulators. 

The Results section describes the results found when exploring the questions shown in section 2 of the application of document standards to each of the three perspectives (i.e., Users, Firms, Regulators). However, the results are subjective and not conclusive. Therefore, I suggest the authors determine which conditions are required for each perspective and show them refined and more understandable.

I suggest a grammar and typos check. For instance, the third paragraph of the Introduction section presents the word "Solidarity", when the right one should be "Solidity". Moreover, in subsection 2.1.5 has the many words wrongly spelled: "persepective", "malfeascence", and "instritutions".

Although the problem's relevance is visible, the paper is premature and needs more development for decisive conclusions.

Author Response

First, thank you for your comments. You were generous in your assessment of the conclusions of my research, and I am pleased the general point of my paper was communicated. However, as I revised the paper, your suggestions went a long way to make that clarity much stronger.

To your individual concerns:

-You asked me to proofread and make sure there were no errors. This was a helpful comment. It did need a good edit and I found several mistakes. Thank you for bringing that to my attention. I believe it is clean now.

-You asked me to reconcile the difference between the document standards and the three perspectives in my methodological description. This was a helpful comment. I don’t believe that these two angles make my methodology any less valid, but I believe there was some ambiguity surrounding that distinction. I inserted a table that bullet points the document standards and conclusions, as well as separates them by each stakeholder’s perspective. It is an extensive chart that I believe really serves to compliment the analysis and to highlight the specific examples that correlate with each audience.

-You mentioned that my results were subjective and inconclusive. This was a great point to bring to my attention. I added some text that explains how my paper is meant simply to point out a way of looking at NFTs or any blockchain record that would need this type of scrutiny with the hope that it would highlight the need for further research in this area. It is to point to an overall need, not come up with conclusive results right now. However, in the table mentioned above, I clearly laid out some more specific examples that would further prove the need for this type of research.

Thank you again. I hope my revisions satisfy your concerns.

Round 2

Reviewer 1 Report

All previous comments have been addressed.

Author Response

thank you 

Reviewer 2 Report

My comments were successfully addressed.

Author Response

thank you